# Global prioritization schemes vary in their impact on the placement of protected areas

**Katie Tjaden-McClement[1], Robin Naidoo[1,2], Angela Brennan[3,4], A. Cole Burton[1,3,5] \***

**1** Department of Forest Resources Management, University of British Columbia, Vancouver, British Columbia, Canada, **2** WWF-US, Washington, District of Columbia, United States of America, **3** Interdisciplinary Biodiversity Solutions Collaboratory, University of British Columbia, Vancouver, Canada, **4** Conservation Science Partners, Inc., Truckee, California, United States of America, **5** Biodiversity Research Centre, University of British Columbia, Vancouver, British Columbia, Canada

\* cole.burton@ubc.ca

## Abstract

In response to global declines in biodiversity, many global conservation prioritization schemes were developed to guide effective protected area establishment. Protected area coverage has grown dramatically since the introduction of several high-profile biodiversity prioritization schemes, but the impact of such schemes on protected area establishment has not been evaluated. We used matching methods and a Before-After Control-Impact causal analysis to evaluate the impact of two key prioritization schemes—Biodiversity Hotspots and Last of the Wild—representing examples of the reactive and proactive ends of the prioritization spectrum. We found that Last of the Wild had a positive impact on the rate of protection in its identified priority areas, but Biodiversity Hotspots did not. Because Biodiversity Hotspots are in or near human-dominated landscapes, this scheme may have been unable to overcome biases towards protecting areas with little human pressure. In contrast, Last of the Wild aligned with the tendency to protect areas far from high human use and thus with lower implementation costs, and so received greater uptake. Stronger links between large-scale prioritizations and more locally driven implementation of area-based conservation, as well as other forms of conservation action, are needed to overcome practical constraints and effectively protect biodiversity on an increasingly human-dominated planet.

## Introduction

Earth's biodiversity is currently under threat due to anthropogenic pressures such as habitat loss and overexploitation, with increasing extinction rates, population declines, and range contractions observed across a wide array of taxa [1]. Establishing protected areas (PAs) has been a cornerstone in efforts to counteract biodiversity declines [2–4]. In addition to their ecological benefits, PAs have been shown to improve well-being for nearby communities [5] and create economic opportunities through ecotourism [6]. In 2010, the Convention on Biological Diversity (CBD) set a goal of protecting at least 17% of ecologically representative terrestrial area globally by 2020 (Aichi Target 11; [7]), motivating dramatic growth in the global PA estate [8], with 17.29% of terrestrial area covered by PAs or other effective area-based conservations

**Data Availability Statement:** All relevant data are publicly available as cited in the manuscript and Supporting information. R code for analysis is available on Zenodo at https://doi.org/10.5281/zenodo.14497727.

**Funding:** The author(s) received no specific funding for this work.

**Competing interests:** The authors have declared that no competing interests exist.

measures (OECMs) as of July 2024 [9]. The CBD's Kunming-Montreal Biodiversity Framework finalized at the COP-15 meeting in December 2022 outlined an updated goal of 30% protection of terrestrial areas by 2030 [10]. This is in line with the 30x30 movement [11] while other organizations advocate for even higher protection targets, like Nature Needs Half, which calls for protecting 50% of the earth by 2030 [12].

Despite the emphasis placed on PAs for biodiversity conservation, they can only be effective if their placement and distribution encompasses the biodiversity they are meant to conserve, and their management promotes the persistence of these species and ecosystems [13, 14]. To meet the challenge of representativeness given limited resources, many prominent environmental non-governmental organizations created global biodiversity prioritization schemes in the late 1990s to early 2000s, mapping the areas they determined to be most important to target for conservation [15]. These schemes can be broadly divided into proactive and reactive approaches, with proactive schemes identifying relatively intact wilderness areas with low human impact and reactive schemes prioritizing areas with high threat levels [15]. Many of these schemes also incorporated a criterion of high irreplaceability (e.g., presence of endemic species), targeting areas that would safeguard the greatest amount of biodiversity for various taxa if conserved [15].

A key motivation for PA prioritization schemes was the realization of important biases and gaps in global biodiversity coverage by PAs. The process of establishing PAs can be demanding and involve biological inventories, stakeholder consultation, infrastructure development, legal designation, and land acquisition, all of which can be costly [13, 16]. Protection can also entail opportunity costs in terms of foregone revenue from resource development [17]. These costs vary across potential sites, typically with higher costs in areas with more human habitation and resource availability or productivity [16]. Thus, the placement of PAs is influenced by economic feasibility, with PAs biased towards areas of "rock and ice" that have fewer conflicting land uses like agriculture or human settlements, but which support less biological diversity [18]. This bias towards areas with low agricultural opportunity cost became more acute over time from 2004 to 2014, and PAs have also not effectively targeted the ranges of threatened vertebrate species [19]. As of 2019, only 21.7% of all threatened species were adequately represented by PAs and only 42.6% of terrestrial ecoregions had met the target of 17% protection [20]. While overall PA coverage in Important Bird Areas and Alliance for Zero Extinction sites increased from 1950 to 2006, the proportion of PA area covering these priority areas relative to non-priority areas has actually decreased over time, globally [21].

For conservation prioritization schemes to impact the placement of new PAs, they must be incorporated into decision-making processes by national and regional governments or other organizations designating PAs. This mechanism of implementation is indirect and requires that decision makers are aware of these schemes and value their utility in identifying areas that will have the greatest impact for conservation. While the CBD Aichi Target 11 required that global PA targets be "important" for biodiversity and ecologically "representative", these metrics were not well defined [22]. Nations committed to meeting these targets may have turned to global prioritization schemes to identify areas for protection that would have the greatest impact for biodiversity conservation, especially in cases where detailed information about biodiversity and conservation priorities at a more local level was not available [23].

If conservation prioritization schemes are effective at improving coverage of biodiversity in PAs, they should lead to reduced biases in coverage of the global PA network. However, this effectiveness has not been rigorously examined. We used a causal inference framework to assess the degree to which two high-profile prioritization schemes, Biodiversity Hotspots (hereafter Hotspots) and Last of the Wild (LOTW), had a positive impact on the placement of protected areas over the past two decades. We focused on these two schemes as prominent

examples of the reactive and proactive ends of the prioritization spectrum. Biodiversity Hotspots is a reactive scheme that prioritized areas based on high levels of both irreplaceability and threat, specifically having >1500 endemic vascular plant species and >70% habitat loss [24]. Conversely, LOTW is a proactive scheme that aimed to protect the world's most pristine wilderness areas by delineating the 10 largest contiguous areas of the 10% "wildest" areas (with the least human footprint) in each biome in each realm [25]. These two schemes are complementary at a global scale, with Hotspots tending to be concentrated in the tropics and in coastal areas, while LOTW has greater coverage in boreal, subtropic, and inland areas (Fig 1). Through this analysis, we do not intend to directly compare Hotspots and LOTW, but rather to investigate these different schemes as examples of reactive and proactive approaches to the challenge of global conservation prioritization.

We tested the hypothesis that the Hotspots and LOTW prioritization schemes positively influenced the creation of PAs, with the associated prediction that the rate of PA growth in

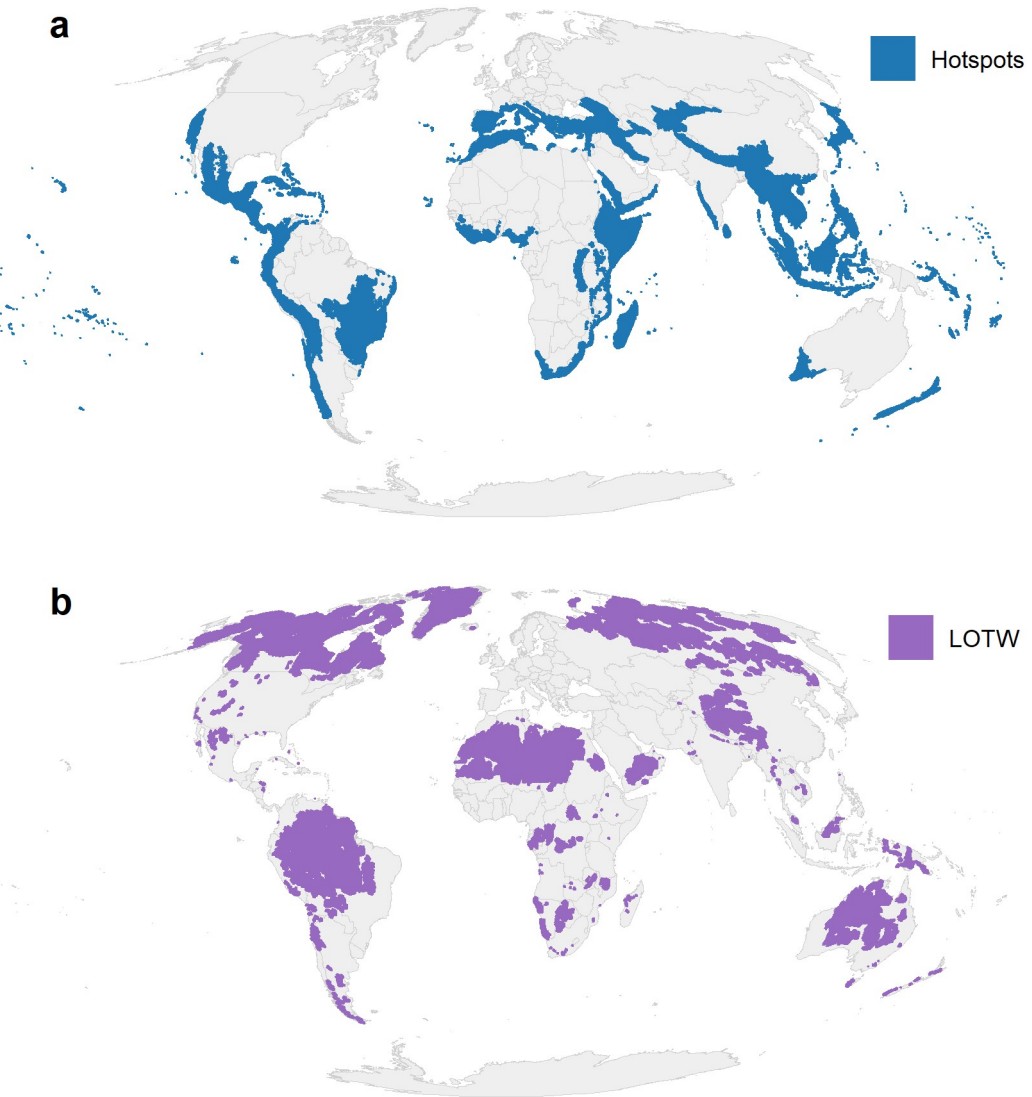

**Fig 1. Priority areas for conserving biodiversity identified by the Biodiversity Hotspots (a; [24, 26]) and Last of the Wild (b; [25]) global prioritization schemes.**

priority areas would be greater than in comparable non-priority areas. We used a statistical matching approach [27] to control for potentially confounding factors between areas inside and outside of prioritization schemes, and a Before-After Control-Impact causal analysis of time-series trends in protection to investigate these hypotheses. While some previous studies have used matching and causal inference techniques to examine other aspects of PA effectiveness [e.g., 28, 29], ours applies these methods to examine the rate of protection in priority areas. Ultimately, our study evaluates the practicality and uptake of global-scale conservation science recommendations within existing policy and governance arenas and can help inform future recommendations about the utility of these types of initiatives and potential barriers to their implementation. Understanding the practicality and limitations of such global initiatives is particularly important as we aim to meet the CBD's new target of protecting 30% of global terrestrial area by 2030.

## Materials and methods

### Data sources and preparation

We used the October 2021 update of the World Database on Protected Areas (WDPA; [30]) to examine the change in spatial extent of protected areas over time within Hotspots and LOTW priority areas (see S1 File for details on pre-processing of the WDPA dataset). We used the equal-area Mollweide projection throughout all spatial analysis to ensure that spatial areas were calculated accurately. Spatial analysis was conducted in R version 4.0.3 [31]. For the two prioritization schemes, we used Version 2 (1995–2004) of the Last of the Wild (LOTW; [25, 32], and the 2004 update of the Biodiversity Hotspots [26]. To process the large spatial datasets, we created a 5 km by 5 km grid of the terrestrial world from the Human Footprint version 3 raster [33] using the "aggregate" function in the "raster" R package [34]. This spatial scale balanced computational efficiency with precision, allowing meaningful insight at a global scale, given an average PA size of 100 km$^2$ [30]. We used a spatial dataset of the centroids of each of these grid cells for subsequent analysis. We then used the "over" function in the "sp" R package [35] to obtain standardized spatial points for the WDPA, Hotspots, and LOTW datasets.

### Matching

We used covariate balancing propensity score (CBPS) matching [36] to select counterfactual "control" groups of 5 km by 5 km terrestrial grid cells (see S1 File for more details) outside of each prioritization scheme. This matching process aimed to replicate the treatment and control groups that would arise from a randomized control trial, with treated sites on average not differing from control sites in relevant, observable variables, except for their designation as a priority area (i.e., treated sites are grid cells within priority areas). This allows stronger causal inference on the impact of establishment of the prioritization schemes on PA placement. Control groups were matched to treatment grid cells on covariates that 1) would impact the likelihood of cells being included in the priority schemes and 2) were likely to affect the outcome variable: the probability of a cell being protected.

   Treatment and control grid cells were matched exactly on country and biome (Terrestrial Ecoregions of the World; [37]) to ensure that matching grid cells came from geographically, politically, and ecologically comparable areas. Within country and biome groups, the matching covariates we used were elevation (Global Multi-resolution Terrain Elevation Data 2010; [38]), human footprint (Global Human Footprint v2 (1995–2004); [32]), agricultural potential [39], human population density (Global Rural-Urban Mapping Project Population Density Grid v1 2000; [40]), and road density (Roads layer from Human Footprint maps circa 2000; [41]; see Table in S1 File for rationale based on previous research and links to data sources for all

matching covariates). Spatial data for the covariates were sourced from the same time period as the establishment of the prioritization schemes (2000 and 2002) to ensure that the matching process created treatment groups that would have had similar potential to be included in those prioritization schemes at that time. While the designation of Hotspots incorporated a criteria of high vascular plant endemism in addition to high human footprint, we did not include this or other measures of biodiversity as matching covariates in our analysis, as such inclusion would have drastically reduced our available sample size for control matches, and these variables are broadly controlled for through our exact matching on country and biome. Furthermore, at least some available evidence suggests that there is little global difference in plant endemism inside vs. outside of protected areas [42].

Matching was performed using the matchit function in the MatchIt R package [43], with calls to the CBPS package for CBPS matching [44]. All matching was done with a 1:1 ratio of treatment to control grid cells, without replacement. Calipers of 0.25 standard deviations on each covariate were added to ensure that matches had similar values to achieve better balance between control and treatment groups [45]. Treatment grid cells with no available matched control cell in the country and biome and within the calipers were dropped from the matched sample (86.32% of Hotspot grid cells, 65.33% of LOTW grid cells dropped).

Matching resulted in datasets of 968,978 total treatment and control grid cells for LOTW (24,224,450 km$^2$) and 256,784 total treatment and control grid cells for Hotspots (6,419,600 km$^2$). We assessed covariate balance between the matched treatment (hereafter referred to as priority) and control groups for each prioritization scheme using the standardized mean differences between groups for each covariate (Fig A in S1 File), examining the distribution of covariates in each group before and after matching (Fig B in S1 File), and through a visual assessment of the plotted group locations (Fig 2). Matching achieved very good covariate balance for both Hotspots and LOTW, with standardized mean differences of less than 0.1 between matched control and priority groups for all covariates, and overall differences less than 0.25, the recommended threshold for regression analyses on matched datasets ([27]; Fig B in in S1 File).

## Causal analysis

We tracked cumulative protected area coverage over time in the matched priority and control groups for each prioritization scheme using the WDPA. We used the year of publication, 2000 for Hotspots [24] and 2002 for LOTW [25], as the time of "establishment" for each scheme. We ran linear models in R to assess the impact of the establishment of each prioritization scheme on the trend in protection using a Before-After Control-Impact (BACI) framework (also referred to as a difference-in-differences design; [46]). These models evaluated the proportion of the total area protected in each matched group of grid cells in a given year as a function of treatment (i.e., priority or control group), time period (i.e., before or after prioritization scheme establishment), year (centered on the year each prioritization scheme was established), and all interactions between these variables [46]. This framework accounts for both immediate and trend differences in protection resulting from prioritization establishment and tests the "parallel trends" assumption—that trends in the control and treatment groups were on a parallel trajectory prior to the treatment intervention (i.e., prioritization establishment)–which is necessary for making causal inference about the treatment. In the model, this "parallel trends" assumption is represented by the interaction between treatment and year. The three-way interaction between treatment, time period, and year represents the impact of prioritization on the trend in protection. We also ran models that tested only for impacts on trend and null models that accounted only for year and treatment group (i.e., no impact of establishment of the prioritization scheme on protected area coverage; Table 1). We compared these alternative models to the full model using AIC$_c$ (Aikaike

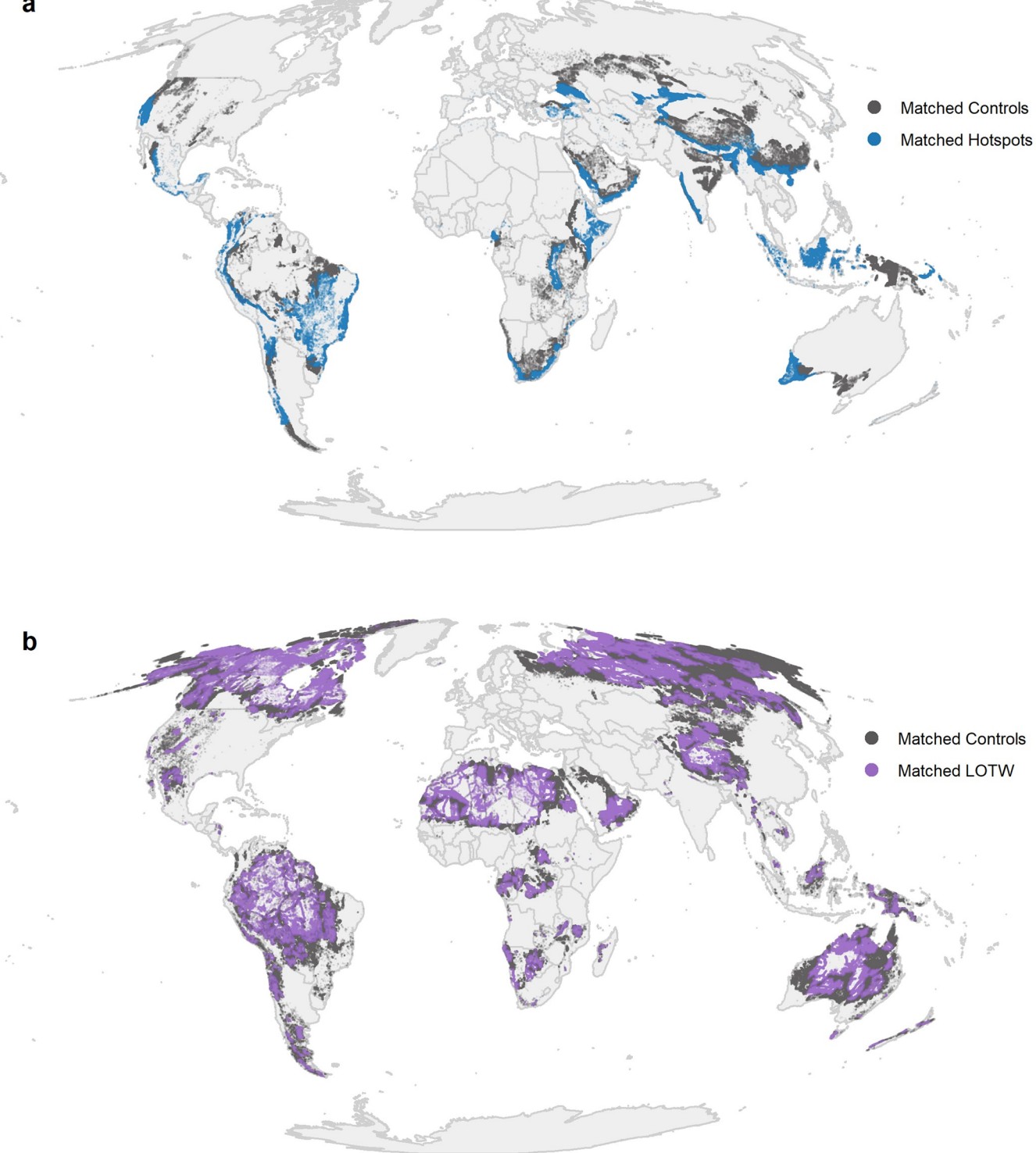

**Fig 2. Global distribution of matched priority and control grid cells for analysis of a) Biodiversity Hotspots and b) Last of the Wild prioritization schemes.**

**Table 1. Model comparison results, based on Akaike Information Criterion (AIC$_c$) values, for Before-After Control-Impact linear models of how two conservation prioritization schemes impacted protected area coverage.** The Full model accounts for impacts of the prioritization scheme establishment on immediate protected area coverage and the trend in protected area coverage; the Trend model only accounts for the impact of the scheme on the trend in protection; and the Null model does not account for any impacts of the establishment of the prioritization scheme on protected area coverage. Number of parameters (K), Akaike Information Criterion adjusted for small sample size (AIC$_c$), differences in AIC$_c$ (ΔAIC$_c$), and AIC$_c$ weights (AIC$_c$Wt), and log-likelihood (logLik).

**a) Biodiversity Hotspots**

| Model | Parameters | K | AIC$_c$ | ΔAIC$_c$ | AIC$_c$Wt | logLik |
|---|---|---|---|---|---|---|
| Full | time period + hotspot + year + time period:hotspot + time period:year + hotspot:year + time period:hotspot:year | 9 | -789.96 | 0.00 | 1.00 | 405.20 |
| Null | hotspot + year | 4 | -741.80 | 48.16 | 3.48e-11 | 375.15 |
| Trend | hotspot + year + time period:year + hotspot:year + time period:hotspot:year | 7 | -739.16 | 50.80 | 9.29e-12 | 377.31 |

**b) Last of the Wild**

| Model | Parameters | K | AIC$_c$ | ΔAIC$_c$ | AIC$_c$Wt | logLik |
|---|---|---|---|---|---|---|
| Full | time period + lotw + year + time period:lotw + time period:year + lotw:year + time period:lotw:year | 9 | -629.64 | 0.00 | 1.00 | 325.04 |
| Trend | lotw + year + time period:year + lotw:year + time period:lotw:year | 7 | -591.76 | 37.88 | 5.95e-09 | 303.62 |
| Null | lotw + year | 4 | -567.01 | 62.64 | 2.51e-14 | 287.76 |

Information Criterion corrected for small sample size bias) to determine if the full model with immediate and trend effects was the best fit to the data [46].

## Results

The full model that included immediate and trend effects was the best supported model (i.e., lowest AIC$_c$) for both conservation prioritization schemes (Table 1). Our results indicated partial support for the hypothesis that prioritization schemes can influence PA coverage. Specifically, LOTW, but not Hotspots, had a positive causal impact on the rate of protected area growth within its designated priority areas.

### Hotspots

The establishment of the Hotspots prioritization scheme had no causal impact on PA coverage within its identified priority areas, relative to comparable control areas. We found no significant differences in the trends in protection after the Hotspots prioritization scheme was established [time period:hotspot:year coefficient estimate (SE) = 0.00018 (0.00015), p = 0.22] or the immediate change in proportion of area protected following the establishment of Hotspots [time period:hotspot coefficient estimate (SE) = 0.00015 (0.0018); p = 0.93; Table 2a]. Protection in the matched Hotspots sample increased from 5.3% to 12.1% area protected from 1980 to 2021, closely tracking the control group which increased from 4.4% to 11.3% (Fig 3a). Both Hotspots and their control group saw an immediate increase in protection after 2000 when the scheme was established, with predicted protection increasing by 0.92% and 0.89% in the Hotspots and control samples, respectively, a jump about six times larger than in other years (Fig 3a). We found that the trends in protection in the matched Hotspots and control groups did not differ significantly prior to the establishment of the Hotspots prioritization scheme in 2000 (hotspot:year coefficient estimate (SE) = -0.000033 (0.00010), p = 0.75; Table 2a), satisfying the parallel trends assumption of the BACI approach.

### Last of the Wild

The establishment of the LOTW scheme resulted in a statistically significant increase in the rate of protection within its priority areas. There was a significant difference in the trends in protection between treatment groups after LOTW was established (time period:lotw:year coefficient estimate (SE) = 0.0014 (0.00039), p = 0.00081; Table 2b), with protection in the LOTW

**Table 2. Model results from linear regressions on time series data of proportion area protected in matched priority and control groups for a) Biodiversity Hotspots and b) Last of the Wild to evaluate each scheme's causal impact on the rate of protection in its identified priority areas.** Time period is whether the prioritization scheme had been established yet, hotspot and lotw are whether the group is the matched priority group (1) or control group (0), and year is the year of the study period (1980 to 2021), centered on 0. Standard errors for each estimated effect size are given.

**a) Biodiversity Hotspots**

| Coefficient | Estimate (SE) | P value |
|---|---|---|
| time period | 0.0075 (0.0013) | < 0.0001*** |
| hotspot | 0.0091 (0.0012) | < 0.0001*** |
| year | 0.0017 (0.000074) | < 0.0001*** |
| time period:hotspot | 0.00015 (0.0018) | 0.93 |
| time period:year | -0.00024 (0.00010) | 0.022 * |
| hotspot:year | -0.000033 (0.00010) | 0.75 |
| time period:hotspot:year | 0.00018 (0.00015) | 0.22 |

**b) Last of the Wild**

| Coefficient | Estimate (SE) | P value |
|---|---|---|
| time period | 0.014 (0.0033) | < 0.0001*** |
| lotw | 0.014 (0.0030) | < 0.0001*** |
| year | 0.0035 (0.00017) | < 0.0001*** |
| time period:lotw | 0.0042 (0.0047) | 0.37224 |
| time period:year | -0.0016 (0.00028) | < 0.0001*** |
| lotw:year | -0.00021 (0.00024) | 0.37940 |
| time period:lotw:year | 0.0014 (0.00039) | 0.00081 *** |

P values are reported with * and *** indicating statistical significance at the $\alpha = 0.05$ and $\alpha = 0.001$ level, respectively.

sample increasing at an estimated rate of 0.14% per year greater than the control group. This increased rate of protection amounts to an estimated additional 322,185 km$^2$ of area protected within the LOTW sample area (12,112,225 km$^2$ total) over the 19 years (2003 to 2021) following the establishment of the LOTW scheme, an area approximately the size of Norway or Vietnam. In the matched LOTW sample, protection increased from 5.2% to 18.6% over the study period, ranging from ~1.5% to 3.5% higher than protection in the matched control group, which increased from 3.6% to 14.9% (Fig 3b). We found an increase in protection after the 2002 establishment of LOTW, with predicted protection increasing by 2.17% and 1.63% in the LOTW and control samples, respectively, a jump roughly 5–9 times that in other years. The trends in the LOTW and control groups prior to 2002 were not found to be significantly different (lotw:year coefficient estimate (SE) = -0.00021 (0.00024), p = 0.38; Table 2b), meeting the parallel trends assumption. There was no significant difference in the immediate changes between the LOTW and control group after LOTW was established (time period:lotw coefficient estimate (SE) = 0.0042 (0.0047); p = 0.37; Table 2b).

## Discussion

We found mixed support for the causal impacts of conservation prioritization on global PA coverage. Specifically, establishment of the Last of the Wild prioritization scheme had a positive impact on the rate of protection in its designated priority areas, but the same was not true for Biodiversity Hotspots. This differing result between two of the most prominent global biodiversity prioritization schemes underscores the fact that prioritization initiatives may not overcome the challenge of conserving areas under high immediate threat. One of the criteria for being prioritized as a Hotspot is a high level of habitat loss, and thus high human pressure

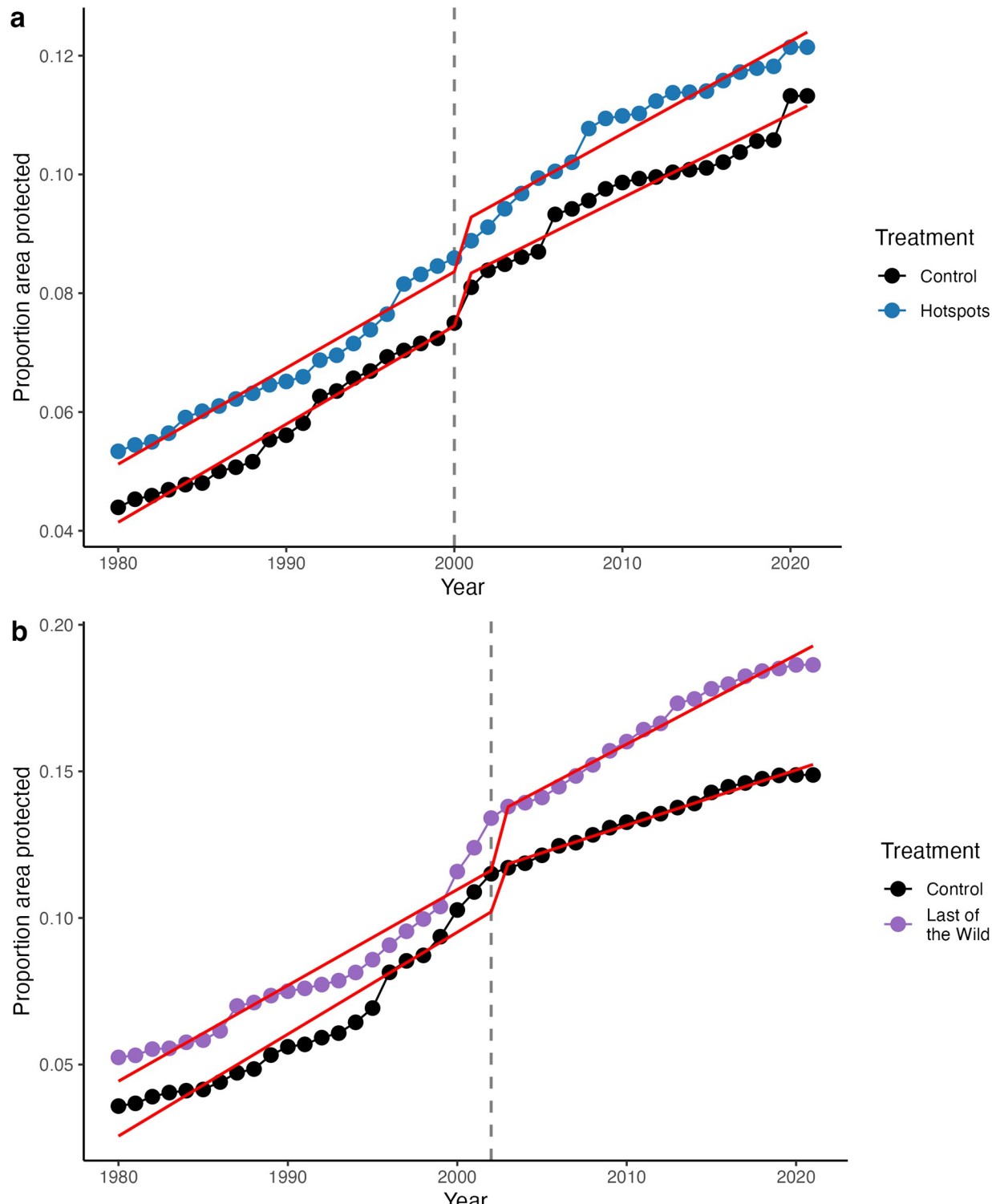

**Fig 3. Trends in proportion of area covered by protected areas from 1980 to 2021 in matched control (black) and priority grid cells for the a) Biodiversity Hotspots (blue) and b) Last of the Wild (purple) prioritization schemes.** Overlayed red trendlines were generated from the linear regression modelling. Dashed grey lines mark the year each prioritization scheme was established.

on natural environments, which inherently makes these areas more difficult to formally protect than those with lower opportunity costs of conservation [47]. When compared to a matched control group similar in human footprint, agricultural potential, population density, and other factors known to influence protection rates, we found that the identification of Hotspots as global priorities for biodiversity protection was not enough to overcome the challenges of implementing protection in these areas valued for human uses. In contrast, LOTW areas were defined based on their low human footprint, making them inherently easier to protect as they face relatively low human pressures. In this case, designating these areas as priorities for conservation resulted in an increase in their rate of protection relative to both their trend in protection before the LOTW scheme was established and a parallel trend observed in a control group with similarly low human footprint.

It is likely that certain Hotspots received greater protection or other conservation investment than non-priority sites, particularly by Conservation International (CI), the organization that developed this prioritization scheme. The Critical Ecosystem Partnership Fund (CEPF), of which CI is a partner, has invested about US$150 million in biodiversity conservation projects in Hotspots since 2001 [48]. However, our results suggest that despite this investment, the overall rate of PA growth across Hotspots was not impacted, relative to control areas. We recognize that area-based protection is only one form of conservation action, and that prioritization schemes may have resulted in other positive conservation outcomes in priority areas (e.g., reduced exploitation, biodiversity-friendly farming). However, given the global and national prominence of PA-based targets, and particularly the recently expanded goals for 2030, we maintain that understanding the impacts of prioritization schemes on global PA placement is important.

## Research limitations and future directions

Our matching criteria ensured that control matches were comparable to priority grid cells on the high-threat criteria for inclusion in the Hotspots scheme and factors likely to influence protection, including country, allowing causal inference. This entailed a trade-off, as matching excluded a large proportion of the priority cells from analysis. Nevertheless, our analysis included samples from 31 of the 34 hotspots, covering over 3 million square kilometers. Another potential limitation of our matching methodology was that we were unable to include plant endemism or other biodiversity variables as matching covariates. We did however select control matches from the same country and biome as the priority samples, which ensures that controls are from broadly similar climate and habitats. Further, if there were a difference, such that PAs are preferentially placed in high-endemism areas, then any bias would be in favor of higher protection in Hotspots, which we did not see. As with all matching studies, we were only able to match on observed variables, leaving open the possibility that unobserved variables (e.g. number of threatened species) could have influenced our results.

Much of the world's terrestrial area has been highlighted as a priority for conservation across the many global prioritization schemes [15, 47, 49]. This provides many options for countries to protect "priority" areas to meet their PA commitments (e.g., protecting 17% of terrestrial area under the Aichi Targets) at a range of price points, but may ultimately dilute the value of "prioritization" for guiding decisions at the scale that PAs are established. Our analysis supports the idea documented elsewhere that governments may have chosen to establish PAs in areas with lower price tags [18, 19]. When global conservation priorities or movements align with what is easiest to protect, they can be effective, but when they do not align, their uptake may be more challenging.

We found that neither prioritization scheme had an immediate impact on the level of protection within its matched priority area. This is likely due in part to a lag in awareness and implementation of the prioritization schemes after their "establishment", which we defined as their year of publication, as well as the time it would take to propose and establish new PAs. Unlike a traditional before-after experimental design, the "treatment" of Hotspot and LOTW areas cannot be cleanly isolated to a single date, so effects are more likely to be observed in trend changes over time as we observed for LOTW, rather than immediate changes. For Hotspots, we used the 2004 revision of the dataset containing 34 hotspots [26], 9 of which were not included in Myers et al. [24]. While rates of protection in these additional hotspots could not have contributed to the immediate effect in our models, they did contribute to the overall trend of protection in Hotspots. We also acknowledge the potential for errors in the WDPA [50], such as the date of establishment for some PAs, but consider it unlikely they would introduce systematic bias to our analysis.

We observed a jump in protection around 2000 (Hotspots) or 2002 (LOTW) for all priority and control groups. This likely reflects a general push to increase the global PA estate at this time. In 2002, 190 countries agreed to the Convention on Biological Diversity's 2010 target of reversing biodiversity loss, with the coverage of PAs serving as an indicator of progress towards this goal [51]. This was also a period of growing awareness of the state of global biodiversity decline due to human impacts [e.g., 52], which prompted many of the global prioritization schemes, but may have also spurred an increase in protection. Further research into the potential impacts of other schemes developed in the same time frame as Hotspots and LOTW, as well as future research on the impacts of recently developed conservation priority layers [e.g., 53] would complement our study.

Global prioritization maps have come under criticism recently for lacking objectivity and having limited utility given the complex local realities of conservation [54, 55]. While we did find a positive impact of the LOTW scheme on PA placement, it is important to recognize the limitations of these types of coarse-scale prioritization mapping efforts. PAs are largely designated by countries or smaller scale jurisdictional authorities, including Indigenous communities [56]. Global maps can provide coarse information where finer scale knowledge is not available, as well as broader context for local decision-making and valuable insight into global progress in safeguarding important areas for biodiversity [23]. Nevertheless, they may provide limited guidance for a national or sub-national government considering where to designate a PA. Many countries either contain no Hotspot or LOTW areas, or the entire country is encompassed by these schemes. This was evident in the matching process, where we found that only about a third of country-biome groups contained both priority and non-priority grid cells. In these cases, more localized conservation planning is likely to be much more relevant and effective. For example, a recent analysis found that the large-landscape scale Yellowstone to Yukon (Y2Y) conservation initiative increased the PA growth rate in the identified region by 90% after it was established in 1993 [57]. Large-landscape conservation programs like Y2Y can consider more localized contexts than global prioritization schemes, are often driven by specific conservation goals (e.g., habitat connectivity for focal species), and are championed by local organizations and individuals, all factors that increase their likelihood of implementation.

## Conclusion

As the world moves into a new set of conservation priorities and targets under the Kunming-Montreal Global Biodiversity Framework [10], it is important to recognize that large-scale prioritizations are only a small part of necessary actions. Ensuring that PAs have adequate

resources to be effectively managed [58], emphasizing the critical role of Indigenous Peoples and other local communities in PA creation and management (e.g., Indigenous Protected and Conserved Areas; [59]), and investing in biodiversity conservation and human-wildlife coexistence outside of PAs, will all be key in moving forward [60–62]. In addition to global prioritization schemes, regional and local conservation initiatives will ultimately determine the success of societies in confronting the critical challenge of conserving biodiversity while supporting human health and well-being.

## Supporting information

**S1 File.**
(DOCX)

## Acknowledgments

We thank members of the Wildlife Coexistence Lab at UBC who provided feedback and support throughout data analysis and manuscript preparation.

## Author Contributions

**Conceptualization:** Katie Tjaden-McClement, Robin Naidoo, A. Cole Burton.

**Data curation:** Katie Tjaden-McClement.

**Formal analysis:** Katie Tjaden-McClement, Angela Brennan.

**Investigation:** Katie Tjaden-McClement, Robin Naidoo, Angela Brennan, A. Cole Burton.

**Methodology:** Katie Tjaden-McClement, Robin Naidoo, Angela Brennan, A. Cole Burton.

**Project administration:** A. Cole Burton.

**Resources:** Robin Naidoo, A. Cole Burton.

**Supervision:** A. Cole Burton.

**Validation:** Angela Brennan.

**Writing – original draft:** Katie Tjaden-McClement.

**Writing – review & editing:** Robin Naidoo, Angela Brennan, A. Cole Burton.

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
