## [Decision Letter · Decision Letter 0]

16 Sep 2024

PONE-D-24-28142Did global prioritization schemes impact the placement of protected areas?PLOS ONE

Dear Dr. Burton,

Thank you for submitting your manuscript to PLOS ONE. After careful consideration, we feel that it has merit but does not fully meet PLOS ONE’s publication criteria as it currently stands. Therefore, we invite you to submit a revised version of the manuscript that addresses the points raised during the review process.

We look forward to receiving your revised manuscript.

Kind regards,

Mattias Gaglio, PhD

Academic Editor

PLOS ONE

Journal Requirements:

3. We note that Figures 1 and 2 in your submission contain [map/satellite] images which may be copyrighted. All PLOS content is published under the Creative Commons Attribution License (CC BY 4.0), which means that the manuscript, images, and Supporting Information files will be freely available online, and any third party is permitted to access, download, copy, distribute, and use these materials in any way, even commercially, with proper attribution. For these reasons, we cannot publish previously copyrighted maps or satellite images created using proprietary data, such as Google software (Google Maps, Street View, and Earth). For more information, see our copyright guidelines: http://journals.plos.org/plosone/s/licenses-and-copyright.

a. You may seek permission from the original copyright holder of Figures 1 and 2 to publish the content specifically under the CC BY 4.0 license.  

Reviewers' comments:

Reviewer's Responses to Questions

**Comments to the Author**

1. Is the manuscript technically sound, and do the data support the conclusions?

Reviewer #1: Yes

Reviewer #2: Yes

Reviewer #3: Yes

2. Has the statistical analysis been performed appropriately and rigorously? 

Reviewer #1: Yes

Reviewer #2: Yes

Reviewer #3: Yes

3. Have the authors made all data underlying the findings in their manuscript fully available?

Reviewer #1: Yes

Reviewer #2: Yes

Reviewer #3: Yes

4. Is the manuscript presented in an intelligible fashion and written in standard English?

Reviewer #1: Yes

Reviewer #2: Yes

Reviewer #3: Yes

5. Review Comments to the Author

Reviewer #1: Overall, the quality of this paper is good.

There should be a conclusion section.

The contributions of the paper should be expand and further highlighted. For example, there are many other matching studies associated with protected areas' conservation status. How does this study differ from existing studies.

The discussion can be restructured to highlight the points being discussed, such as a seperate sub-section of research limitation; a sub-section of suggestions for further research; a sub-section of policy implications.

Reviewer #2: In this study, the authors conduct a much-needed assessment of the effectiveness of two conservation prioritisation schemes, i.e., LOTW and BHs, when protecting biodiversity throughout key priority areas. Overall, this study is scientifically sound and exceptionally well-written. I would like to extend my thanks to the authorship team for their dedication to an important research topic. I only have relatively minor comments for the authors to consider, but by no means expect them to include them all in a revised manuscript.

Specifically, the role of PAs in generating ecotourism opportunities and thus income for local, rural or indigenous communities has not been mentioned in the introduction, I suggest including a sentence or two on this to provide some context on the wider benefits of PAs beyond just conserving biodiversity. Can the authors expand on the implementation process of PAs in relation to their economic costs, as this concept has been touched on in the abstract and discussion but not adequately explained in enough detail for readers unfamiliar with the process to understand, i.e., what makes a particular site more/or less costly to protect?

Please see below for more detailed comments.

All the best,

Connor T. Panter

Detailed comments:

Title – while I am not against titles featuring questions, I think that this title could be improved by specifically stating the main takeaway message here. Something along the lines of “Last of the Wild protected areas, but not Biodiversity Hotspots, are effective at protecting global biodiversity in key priority areas”.

Line 33 – the term “cheaper” here is a little confusing without relevant clarification. Either expand on what you mean here or use alternative phrasing.

Line 53 – E.O. Wilson’s book Half-Earth springs to mind here.

Line 56 – management also includes commitment to protect and enforcement/prosecution by bad actors by regional authorities, too. Perhaps expand on this a little.

Line 132 – what spatial resolution were the PA data downloaded in?

Line 132 – why did the authors decide on 5 x 5 km grid cells? What was the justification for this? I ask because I have the IUCN Red List Area of Occupancy 2 x 2 km resolution in mind… understand that this could be due to computing power, etc. Perhaps some extra clarification on the decision-making process here is needed.

Table 1 – not sure whether scientific notation for 1.00e+00 is necessary here?

Line 250 – again reconsider the use of scientific notation in text here, I don’t see the issue with writing out the value in full here?

Table 2 – P value column, why not just present very small values as “< 0.0001”? same for model estimates (SE).

Line 335 – avoid contractions, i.e., “did not”.

Line 336 – such as what? Can you provide some examples of unobserved variables for context here?

Line 346 – contraction.

Line 359 – I agree.

Line 368 – reconsider the word “contemporaneously” …

Line 376 – contraction.

Reviewer #3: One of the most importat conservation actions challenges on this days are the prioritization for decision makers. This papper present another way to approch this question.

This manuscript is technically clear, also the statistical analysis is decribed and can be replicable and all data is available.

- Lines 78 to 82 is in the introduction part but without references, goes better on the discussion part.

- Lines 313 to 318

313 It is likely that certain Hotspots received greater protection or other conservation

314 investment than non-priority sites, particularly by Conservation International (CI), the

315 organization that developed this prioritization scheme. The Critical Ecosystem Partnership Fund

316 (CEPF), of which CI is a partner, has invested about US$150 million in biodiversity conservation

317 projects in Hotspots since 2001 (CEPF, 2021). However, our results suggest that this investment

318 did not impact the rate of formally protecting Hotspots as a whole, relative to control areas.

Please explain better this, seems like is direct correlate the financial investment to the results, but it can be other factors.

- Line 344 reference Joppa & Pfaff 2008, its not listed on references.

- Line 354 reference Mittermier et al, with no year I assume that is MitermEier et al 2004, please correct.

- Line 462 reference Robert J. Hijmans; other references are first the last name and only the initials of the name.

6. PLOS authors have the option to publish the peer review history of their article (what does this mean?). If published, this will include your full peer review and any attached files.

Reviewer #1: No

Reviewer #2: No

Reviewer #3: **Yes: **María Félix-Lizárraga

---

## [Author Response · Author response to Decision Letter 0]

20 Nov 2024

Please find all of our responses and revisions detailed in the Response to Reviewers document (and tracked changes version of manuscript)

---

## [Decision Letter · Decision Letter 1]

9 Dec 2024

Global prioritization schemes vary in their impact on the placement of protected areas

PONE-D-24-28142R1

Dear Dr. Burton,

We’re pleased to inform you that your manuscript has been judged scientifically suitable for publication and will be formally accepted for publication once it meets all outstanding technical requirements.

Kind regards,

Mattias Gaglio, PhD

Academic Editor

PLOS ONE

Additional Editor Comments (optional):

Reviewers' comments:

Reviewer's Responses to Questions

**Comments to the Author**

1. If the authors have adequately addressed your comments raised in a previous round of review and you feel that this manuscript is now acceptable for publication, you may indicate that here to bypass the “Comments to the Author” section, enter your conflict of interest statement in the “Confidential to Editor” section, and submit your "Accept" recommendation.

Reviewer #2: All comments have been addressed

Reviewer #3: All comments have been addressed

2. Is the manuscript technically sound, and do the data support the conclusions?

Reviewer #2: (No Response)

Reviewer #3: Yes

3. Has the statistical analysis been performed appropriately and rigorously? 

Reviewer #2: (No Response)

Reviewer #3: Yes

4. Have the authors made all data underlying the findings in their manuscript fully available?

Reviewer #2: (No Response)

Reviewer #3: Yes

5. Is the manuscript presented in an intelligible fashion and written in standard English?

Reviewer #2: (No Response)

Reviewer #3: Yes

6. Review Comments to the Author

Reviewer #2: (No Response)

Reviewer #3: The authors address the comments, or explain them-self better. I don´t have any concerns about dual publication, research or publication ethics.

7. PLOS authors have the option to publish the peer review history of their article (what does this mean?). If published, this will include your full peer review and any attached files.

Reviewer #2: No

Reviewer #3: No

---

## [Editor Report · Acceptance letter]

18 Dec 2024

PONE-D-24-28142R1 

PLOS ONE

Dear Dr. Burton, 

I'm pleased to inform you that your manuscript has been deemed suitable for publication in PLOS ONE. Congratulations! Your manuscript is now being handed over to our production team.

Kind regards, 

on behalf of

Dr. Mattias Gaglio 

Academic Editor

PLOS ONE